# The superficial layers of the primary visual cortex create a saliency map that feeds forward to the parietal cortex

Chen Liu[1,2☉], Chengwen Liu[3☉], Laurentius Huber[4], Li Zhaoping[5], Peng Zhang [1,2*]

**1** State Key Laboratory of Cognitive Science and Mental Health, Institute of Biophysics, Chinese Academy of Sciences, Beijing, China, **2** College of Life Sciences, University of Chinese Academy of Sciences, Beijing, China, **3** Institute of Interdisciplinary Studies, Cognition and Human Behavior Key Laboratory of Hunan Province, School of Educational Science, Hunan Normal University, Changsha, China, **4** Functional MRI Core, National Institute of Mental Health, National Institutes of Health, Bethesda, Maryland, United States of America, **5** University of Tübingen, Max Planck Institute for Biological Cybernetics, Tübingen, Germany

☉ These authors contributed equally to this work.
* zhangpeng@ibp.ac.cn

## Abstract

A salient visual object with a distinct feature from the surrounding environment automatically captures attention. While the saliency signals have been found in many brain regions, their source remains highly controversial. Here, we investigated the neural origin of visual saliency using cortical layer-dependent functional magnetic resonance imaging (fMRI) of cerebral blood volume (CBV) at 7 Tesla. Behaviorally, human observers were better at detecting salient foreground bars with a larger orientation contrast from uniformly oriented background bars. Saliency-sensitive signals were strongest in the superficial layers of the primary visual cortex (V1) and in the middle layers of the intraparietal sulcus (IPS) of the parietal cortex. Layer-dependent effective connectivity revealed the transmission of saliency signals along the feedforward pathway from V1 to IPS. Furthermore, behavioral sensitivity to the foreground stimulus correlated significantly with the fMRI response in the superficial layers of V1. Our findings provide mesoscale evidence that a visual saliency map is created by iso-feature suppression through lateral inhibition in the superficial layers of V1, and then feeds forward to attentional control brain regions to guide attention and eye movements.

## Introduction

Only a small fraction of sensory inputs can be selected by our attention for further processing. Attention is guided both endogenously by top–down factors and exogenously by bottom-up factors [1–5]. Although the bottom-up guidance is simpler, it remains controversial which brain area computes saliency from exogenous visual

**Data availability statement:** Data to reproduce the main findings of this study can be downloaded from National Basic Science Data Center (https://www.scidb.cn/doi/10.1101/2025.04.10.648136). The mripy package, used in this study for high-resolution fMRI data processing, is available on Github (https://github.com/herrlich10/mripy). A permanent copy of the current version can be found in mripy-copy.zip at https://www.scidb.cn/doi/10.1101/2025.04.10.648136.

**Funding:** This study was supported by Ministry of Science and Technology of China (https://en.most.gov.cn/) STI2030-Major Projects (2022ZD0211900 and 2022ZD0204200 to P.Z.), National Natural Science Foundation of China (https://www.nsfc.gov.cn/english/site_1/index.html, 31871107 and 31930053 to P.Z.), Hunan Provincial Natural Science Foundation (2024JJ6313 to C.W.L), Scientific Research Foundation of Hunan Provincial Education Bureau (24B0058 to C.W.L) NIH Intramural Program of NIMH/NINDS (#ZIC MH002884 to L.H.), the Max Planck Society and the University of Tübingen. The funders had no role in study design, data collection and analysis, decision to publish, or preparation of the manuscript.

**Competing interests:** The authors declare no competing interests.

**Abbreviations:** BCEA, bivariate contour ellipse area; BOLD, blood oxygen level dependent; CSF, cerebrospinal fluid; CBV, cerebral blood volume; fMRI, functional magnetic resonance imaging; FWE, family-wise error; GLM, general linear model; GM, gray matter; gPPI, generalized psychophysiological interaction; IPS, intraparietal sulcus; pRF, population receptive field; rm, repeated-measures; ROIs, regions of interest; SC, superior colliculus; SNR, signal-to-noise ratio; VASO, Vascular Space Occupancy; V1SH, V1 Saliency Hypothesis; WM, white matter.

input. It is believed that the brain generates a saliency map from feature contrast to guide attention shifts and eye movements [6–9]. According to a traditional and popular computational model [7,10,11], different feature maps (e.g., color, luminance, orientation, etc.) are generated by center-surround contrast mechanisms and then combined into a feature-agnostic master map of saliency, presumably in a high-order brain region such as the parietal cortex. However, it has been shown that saliency behavior can also be explained by the V1 Saliency Hypothesis (V1SH) [8,12], according to which the saliency map is generated by intracortical mechanisms in the primary visual cortex (V1) and represented by the firing rates of feature-selective neurons.

Visual saliency-related signals have been observed in the frontal cortices [13,14], parietal cortices [2,15], V4 [16], V1 [17,18], as well as subcortical regions including the pulvinar of the thalamus [19] and the superior colliculus (SC) of the midbrain [20–22]. An important question is which area generates the saliency signals originally and which areas merely receive and utilize the saliency signals generated in another area. For example, a recent study showed that inactivation of the posterior parietal cortex in monkeys significantly weakened the relationship between image salience and visually guided behavior [15], this could suggest that the parietal cortex is the original source of the saliency signals, but this is also consistent with the idea that parietal cortex utilizes the saliency signals and combine them with top–down signals for guiding behavior [2]. Using salient visual stimuli that were made subliminal by backward masking, a human neuroimaging study found responses to saliency in the earliest (C1) component of event-related potentials, and functional magnetic resonance imaging (fMRI) responses to saliency in V1 but not in the parietal cortex [18]. However, backward masking may not fully abolish top–down attention, since attention can modulates stimulus processing even without awareness [23,24] and C1 amplitude [25–27]. Meanwhile, supraliminal inputs do evoke parietal responses, as demonstrated by both fMRI and neurophysiological studies [28,29]. Hence, the lack of a saliency signal in IPS could be due to the limited spatial resolution of 3T fMRI to isolate the retinotopic signal corresponding to the spatially localized salient location. Furthermore, a monkey electrophysiological study reported that the latency of the saliency signals in SC is shorter than that in V1 [22], but this short SC latency is longer than the latency of the saliency signals in V1 reported in another monkey study [17]. Therefore, neuroscience studies have found saliency-related signals in multiple brain regions, and it remains controversial as to which brain regions are, respectively, the sources and recipients of the saliency signals. In this study, we aim to shed light on the origin of saliency signals through high-resolution fMRI, which allows us to examine laminar patterns of responses to saliency signals. As will be explained later, the laminar patterns are diagnostic of the sources of the saliency signals, and we will focus on the question of whether the source for the saliency signals is in V1, IPS, or SC.

Recent advances in high-resolution fMRI at ultra-high magnetic field have enabled noninvasive imaging of mesoscale functional units, such as cortical layers and columns, in the human brain at submillimeter resolution [30–32]. Using submillimeter blood oxygen level dependent (BOLD) fMRI techniques, feedforward and feedback

pathways of stimulus-driven activity and top–down attentional modulation have been investigated in the human visual cortex [33–36]. However, the laminar specificity of T2* weighted BOLD fMRI is limited by the draining vein effect [37–40]. The Vascular Space Occupancy (VASO)-based cerebral blood volume (CBV) technique is an inversion recovery MRI technique that detects changes in blood volume by nulling the signal from blood water [41,42], particularly in small vessels such as arterioles and capillaries. Therefore, compared to T2*w BOLD signals that are sensitive to blood oxygen in large veins, CBV changes are more closely related to the site of neuronal activation, offering higher spatial specificity to resolve layer-dependent signals across cortical depth, despite having relatively lower functional sensitivity [43,44].

Using laminar fMRI techniques, one can examine the laminar patterns of responses to visual saliency in various brain regions to diagnose the source of the saliency signals. Fig 1 schematizes the laminar patterns of saliency-related signals in V1 and parietal cortex according to each of the three possible origins of the saliency signal: parietal cortex, V1, and SC, based on the canonical neural circuits in the primate visual system [45–48]. If the parietal cortex generates the saliency map and modulates V1 activity through feedback connections, the laminar pattern of saliency-related signals should be consistent with cortico-cortical feedback, stronger in the deep and superficial layers of V1 and in the deep layer of the parietal cortex. If, according to the V1SH, saliency signals emerge in V1 through intracortical interactions mediated by horizontal connections in the superficial layers of V1, the strongest saliency-related activity should be observed in the superficial layers of V1 and in the middle layers of high-level cortex that receive feedforward input. Finally, if the saliency signals originate in the SC, they would modulate activity more strongly in the middle and superficial cortical layers through the tecto-thalamo-cortical pathway [19,48,49]. Specifically, this pathway may relay through the pulvinar, terminating in the middle and superficial cortical layers, or through the LGN, which primarily targets the middle layers of V1. Overall, the middle layers receive stronger thalamic input than the superficial layers [49]. Thus, although the SC was not recorded in the current study due to the limited field-of-view of VASO fMRI, the laminar profiles of saliency signals recorded in the cortical regions can still be used to test the hypothesis whether these signals originate from a subcortical source.

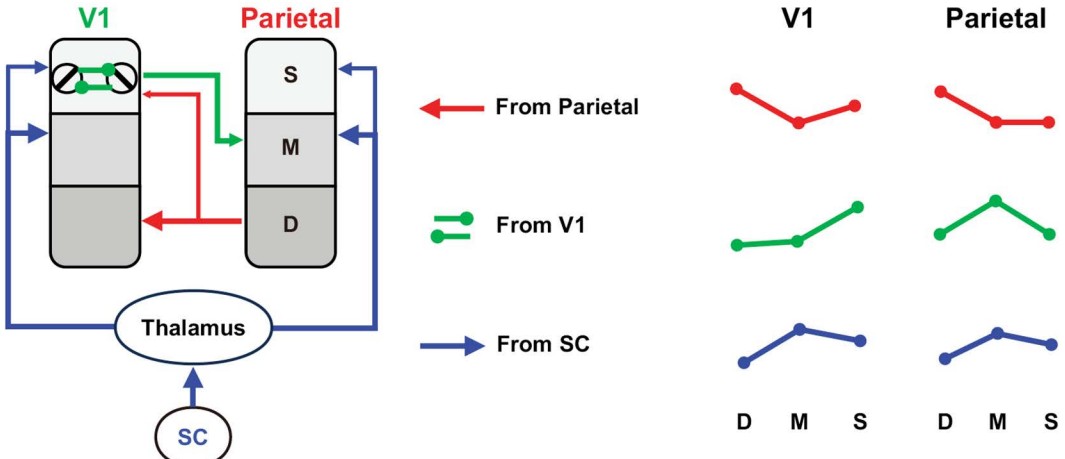

**Fig 1. Three possible hypotheses on the origins of saliency signals and their consequent laminar patterns of the saliency signals through feedforward and feedback neural pathways.** In the feature combination hypothesis, saliency signal initially emerged in frontoparietal areas and then modulate neural activity of parietal and early visual cortices through feedback connections (Red lines); In V1SH, saliency signals initially emerge via intracortical interactions mediated by the horizontal connections in superficial layers of V1 (primary visual cortex). These signals can then feed forward to middle layers of parietal cortex (Green lines); In the subcortical computation hypothesis, saliency signals are generated in superior colliculus (SC) and then modulate middle and superficial layer activity of cerebral cortex through thalamo-cortical pathways. On the right, relative strengths of saliency signals in the deep (D), middle (M), and superficial (S) layers of V1 and parietal cortex that should arise from each hypothesis are shown. These indicated relative strengths are schematic representations derived from existing literature on the hierarchical connections of brain regions.

In the current study, to reveal the neural source of visual saliency, we employed submillimeter fMRI at 7 Tesla with a CBV-weighted VASO fMRI technique to investigate cortical depth-dependent responses to a foreground stimulus consisting of four bars presented at different orientation contrasts ($\theta = 90°$, 15°, or 0°) relative to a background of iso-oriented bars (Fig 2). To link the fMRI responses with the bottom-up saliency of the foreground region rather than top–down task goals, the fMRI responses were collected from participants who were performing a demanding fixation task at a central fixation cross away from the foreground region. Consequently, the peripheral texture stimuli, including the foreground region, were

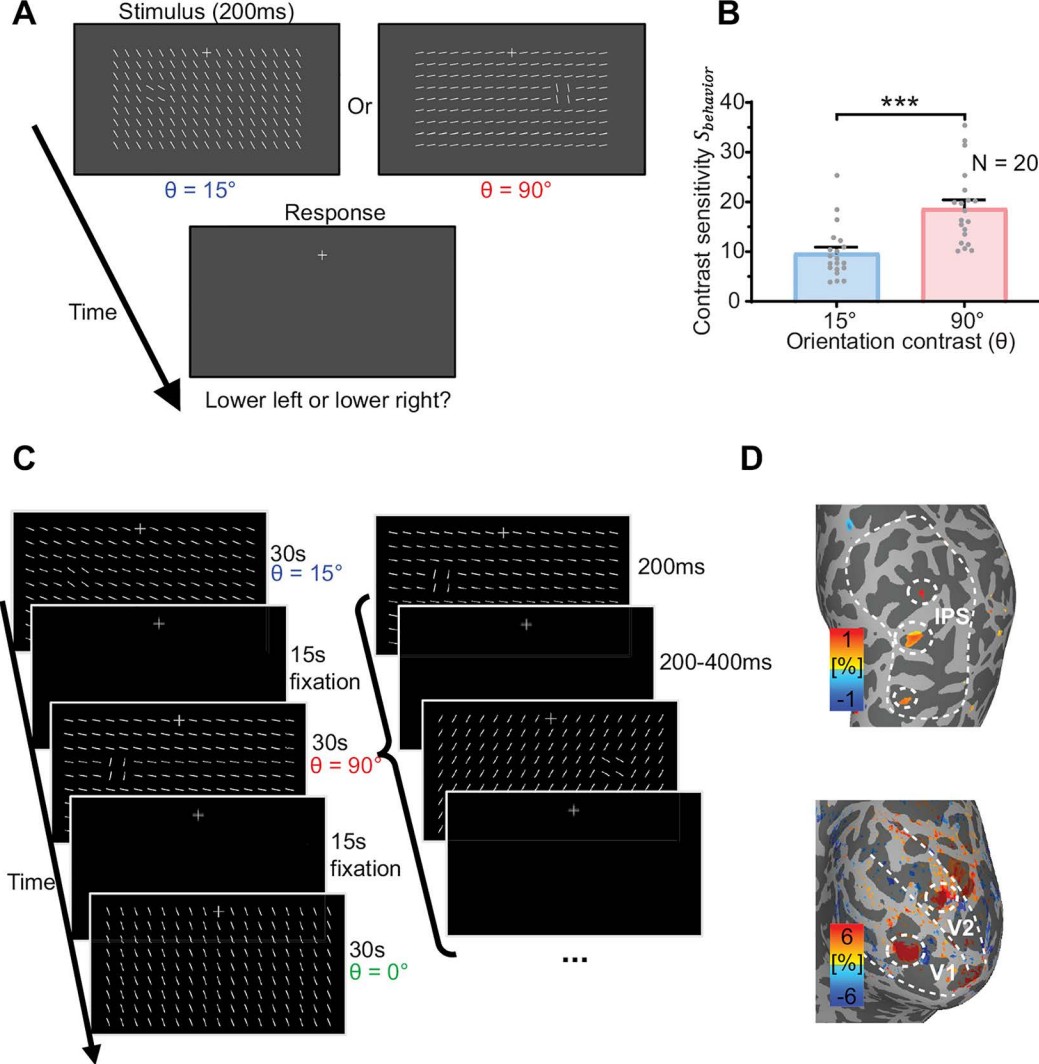

**Fig 2. Stimuli and procedures for the behavioral and fMRI experiments.** (A) Stimulus and procedure of the psychophysical experiment outside the MRI scanner. The orientation contrast between foreground and background bars is defined as θ. (B) Luminance contrast sensitivity ($S_{behavior}$) for detecting the 90° and 15° foreground. Each dot represents data from one participant. Error bars represent standard errors of the mean (SEM). *** $p < 0.001$. (C) Stimulus and procedure of the fMRI experiment. Texture stimuli with three θ conditions were presented in separate blocks. The foreground bars were randomly presented to the lower-left or lower-right of fixation in each block while participants performed a central fixation task. (D) Localizer activations to the retinotopic location of the foreground bars in V1 and V2 of the early visual cortex and IPS of the parietal cortex (thresholded at $p < 0.05$ uncorrected in a representative participant), as indicated by the dashed circles. Dashed ellipses enclose the regions of interests to the foreground region in the contralateral visual field. Dashed lines mark the borders of the brain areas. The data underlying this Figure can be found in *data/main_data.xlsx* at https://www.scidb.cn/doi/10.1101/2025.04.10.648136.

task-irrelevant. The laminar profile of the saliency-sensitive fMRI response, its correlation with behavioral sensitivity to saliency, and the layer-dependent functional connectivity collectively support the hypothesis that the saliency map is generated in the superficial layers of V1 and subsequently projects to attention-control brain regions, such as the parietal cortex.

## Results

### Behavioral sensitivity was higher to a foreground region with a larger orientation contrast against the background

In a separate psychophysical experiment outside the MRI scanner, we measured participants' behavioral sensitivity to detect foreground bars with different orientation contrasts against the iso-oriented background bars. In each trial, the texture stimulus, containing the foreground and background bars, was presented for 200 ms (Fig 2A). The foreground bars were presented either to the lower-left or the lower-right of the fixation, at an eccentricity of 4.3°, and participants pressed one of two buttons to report whether the foreground bars were at lower-left or lower right. The orientation of the uniformly oriented background bars was randomly chosen for each trial, while the foreground bars had an orientation contrast $\theta$ of either 90° or 15° relative to the background bars. Across trials, the Michaelson luminance contrast between the bars (all of which had the same luminance) and the uniform gray background was adaptively adjusted using a 3-down-1-up staircase procedure, implemented independently for the two $\theta$ conditions, to identify a threshold contrast $C_{threshold}$ at which the participant can perform the task at 80% accuracy. The results showed that luminance contrast sensitivity $S_{behavior} = 1/C_{threshold}$ was significantly higher for a foreground with a larger orientation contrast $\theta$ (Fig 2B, 90° versus 15°, $t_{19} = 13.829$, $p < 0.001$). Specifically, the behavioral sensitivity to detect the 90° foreground ($S_{behavior}(90°) = 18.719$) was approximately twice that of the sensitivity to detect the 15° foreground ($S_{behavior}(15°) = 9.720$) (Fig 2B). It should be noted that, at our luminance contrast sensitivity $S_{behavior}$ to detect the foreground, the luminance of all the bars were presented at a suprathreshold level. Hence, the luminance contrast threshold $C_{threshold}$ measured in our task does not reflect the visibility of the entire stimulus but rather the saliency of the foreground bars in the context of the background texture. When the orientation contrast $\theta$ or saliency is higher, the foreground is more easily detected and localized, so that our participants could do our task at a lower luminance contrast of the bars. Hence, our luminance contrast sensitivity $S_{behavior}$ is an assessment of the saliency of the foreground bars, and we also call $S_{behavior}$ our *behavioral saliency score*. Furthermore, to correlate with the fMRI signals for saliency later, we define

$$behavioral\ sensitivity\ to\ saliency, \quad \text{QUOTE} \quad SS_{behavior} = (S_{behavior}(90°) - S_{behavior}(15°))/mean(S_{behavior}).$$

Here, $mean(S_{behavior})$ is the average of $S_{behavior}(90°)$ and $S_{behavior}(15°)$.

### Saliency-sensitive responses were strongest in the superficial layers of V1 and in the middle layers of IPS

In the 7T fMRI experiment, a slice-saturation slab-inversion VASO (SS-SI-VASO) sequence with 0.82-mm isotropic resolution was used to measure CBV-weighted signals across cortical depth from the posterior part of the brain [44,50]. In each inversion TR, blood-nulled and not-nulled 3D-EPI volumes were acquired to minimize T2*-weighted BOLD contamination. The imaging slab was oriented in an oblique-coronal orientation to cover the early visual cortices and the parietal cortex from the dorsal part of the occipito-parietal lobe (S1C Fig). According to the literature and hypotheses on the neural origin of visual saliency (Fig 1), our analyses focused on V1 and V2 of the early visual cortex, and the intraparietal sulcus (IPS) of the parietal cortex. As an intermediate level in the cortical hierarchy between V1 and IPS, V2 was expected to exhibit a laminar profile in-between those of the two brain regions. Thus, the laminar pattern of V2 can provide further information to distinguish the laminar hypotheses schematized in Fig 1.

The stimulus and procedure of the fMRI experiment are illustrated in Fig 2C. Bar stimuli with three orientation contrast conditions ($\theta = 90°$, 15°, and 0°, randomly clockwise or counterclockwise to the orientation of the background bars) were

presented in the lower visual field in separate 30-s blocks, interleaved with 15-s fixation period. Within a stimulus block, $\theta$ is fixed, and each stimulus was presented for 200 ms, with a random inter-stimulus interval ranging from 200 to 400 ms. In each stimulus, the iso-oriented background bars had a random orientation, and the foreground bars appeared randomly either to the lower left or lower right of the fixation. During the experiment, participants were instructed to maintain fixation and to count the number of luminance changes of the fixation cross. The fixation task was designed to be attentionally demanding by the subtlety and the large number of the luminance changes in each run. Consequently, the reported number of luminance changes was correct in only 32.0% of the runs, and deviated from the correct number by 4.2% on average. The performance indicates that the task was sufficiently challenging and yet participants remained engaged and performed well (S2 Fig). Furthermore, eye-tracking data confirmed that participants maintained stable fixation without systematically biased toward the foreground stimuli (S3 Fig).

In two localizer fMRI runs, naturalistic stimuli of the same size and location as the foreground region were presented in the lower-left and lower-right visual fields in separate blocks while participants performed a same fixation task (S4 Fig). The fMRI responses to the natural stimuli were used to identify the cortical regions of interest (ROIs) for the foreground bars, as shown in Fig 2D for a representative participant.

To quantify our results and to account for individual differences in mean response amplitude, we define the normalized CBV foreground response as $S_{fMRI}(s, l, \theta) = \frac{r(s, l, \theta)}{r_{norm}(s)} \bar{r}$. In which, $r(s, l, \theta)$ is the original CBV response for subject $s$, layer $l$, at foreground orientation contrast $\theta$, $r_{norm}(s) = \sqrt{\frac{\sum_{l,\theta} r(s,l,\theta)^2}{\sum_{l,\theta} 1}}$ and $\bar{r} = \frac{\sum_s r_{norm}(s)}{\sum_s 1}$. CBV-weighted fMRI responses $S_{fMRI}$ in the foreground ROIs in V1, V2, and IPS are plotted as a function of cortical depth in Fig 3A (see S5 Fig for the BOLD response profiles and S6 Fig for VASO response profiles extended into the white matter [WM] and cerebrospinal fluid [CSF]).

A two-way repeated-measures (rm) ANOVA with cortical depth (deep, middle, and superficial) and orientation contrast ($\theta = 90°$, 15°, and 0°) as within-subject factors was performed on the normalized foreground response $S_{fMRI}$. A significant effect of orientation contrast $\theta$ was found in all three brain regions (V1: $F_{2, 38} = 7.080$, p = 0.002; V2: $F_{2, 38} = 4.763$, $p = 0.014$; IPS: $F_{2, 38} = 4.673$, $p = 0.015$), indicating stronger responses to the foreground with larger orientation contrast (i.e., 90° compared to 15° and 0° conditions). However, CBV responses to the 15° foreground were slightly reduced compared to the 0° condition. This is due to the partial volume effect of background suppression (S7 Fig). Similar effects of background suppression were found for the 90° and 15° conditions. Therefore, to avoid the partial volume effect of background suppression, *the saliency-sensitive fMRI response $SS_{fMRI} = S_{fMRI}(90°) - S_{fMRI}(15°)$* was calculated as the difference between the fMRI responses to the 90° foreground and the 15° foreground conditions (Fig 3A, lower).

The amplitude of $SS_{fMRI}$ differed significantly across cortical depth in V1, V2, and IPS (V1: $F_{2, 38} = 9.696$, p = 0.002; V2: $F_{2, 38} = 9.833$, p = 0.001; IPS: $F_{2, 38} = 5.421$, p = 0.008). A significant interaction effect was found between depth and ROI, indicating that the effect of the cortical depth on the $SS_{fMRI}$ depended on the ROIs ($F_{4, 76} = 6.052$, p = 0.002). In V1, the saliency-sensitive response was strongest in the superficial layers (S versus D: $t_{19} = 3.408$, p = 0.003; S versus M: $t_{19} = 3.660$, p = 0.002). In V2, it was stronger in the superficial and middle layers than in the deep layers (S versus D: $t_{19} = 3.232$, p = 0.004; M versus D: $t_{19} = 4.582$, p < 0.001). In IPS, $SS_{fMRI}$ was strongest in the middle layers (M versus D: $t_{19} = 2.659$, p = 0.015; M versus S: $t_{19} = 2.983$, p = 0.008). Saliency-sensitive responses calculated by the original CBV responses showed a similar pattern of results (S8 Fig). The laminar profiles of the response difference between the 90° and 0° conditions, $S_{fMRI}(90°) - S_{fMRI}(0°)$, were also qualitatively similar (S9 Fig). To account for differences in signal-to-noise ratio (SNR) across cortical layers in the observed laminar profile, we normalized the saliency-sensitive responses $S_{fMRI}$ by the SNR at each cortical depth. The normalized responses still exhibited similar laminar profiles (S10 Fig). In Fig 3B, the activation map in a representative participant clearly showed the strongest saliency sensitivity $SS_{fMRI}$ in the superficial layers of V1, in the middle and superficial layers of V2, and in the middle layers of IPS, consistent with the laminar profiles for a V1 origin of saliency in Fig 3A. To better illustrate the V1 saliency map across cortical depths, saliency-sensitive responses $SS_{fMRI}$ were shown on the cortical surface in the right hemisphere of the representative subject

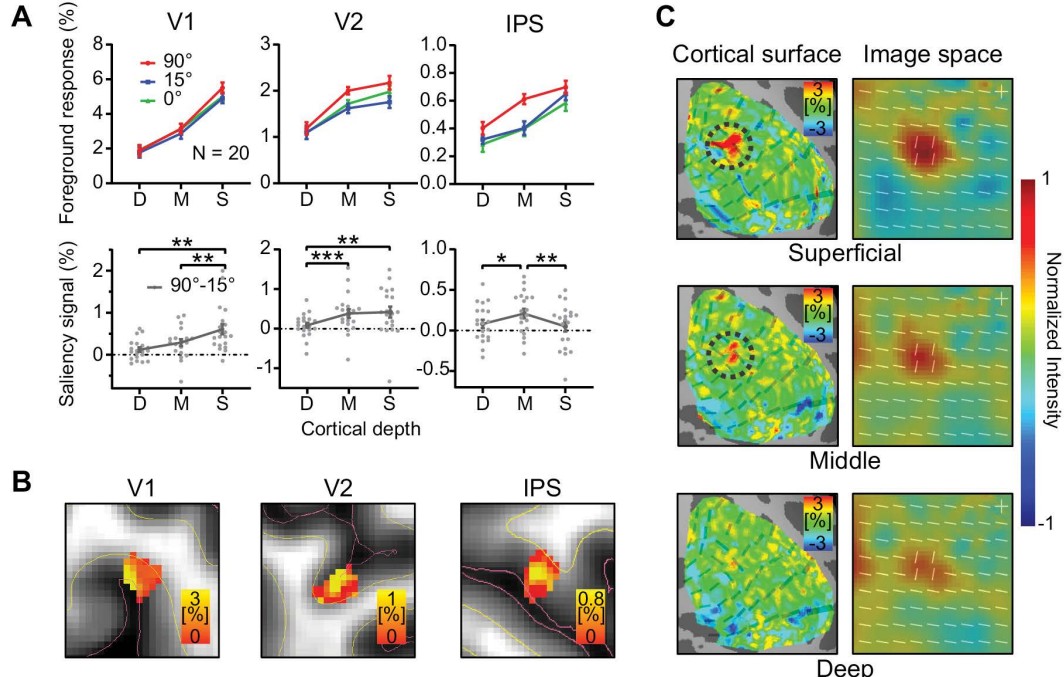

**Fig 3. Laminar fMRI responses in V1, V2 and IPS.** (A) Upper: Normalized CBV responses ($S_{fMRI}$ in percent signal change) to the orientation fore-grounds in V1, V2, and IPS. Lower: CBV response difference between the 90° and 15° foreground conditions ($SS_{fMRI} = S_{fMRI}(90°) - S_{fMRI}(15°)$). Each gray dot represents data from one participant. Error bars represent SEM. *, ** and *** indicate $p < 0.05$, $p < 0.01$, and $p < 0.001$, respectively. D: deep, M: middle, S: superficial. **(B)** Volume activation maps of the saliency-sensitive response $SS_{fMRI}$ in the foreground ROIs of V1, V2, and IPS for a representative participant. Red lines indicate the boundary between gray matter (GM) and cerebrospinal fluid (CSF). Yellow lines indicate the bound-ary between GM and white matter (WM). **(C)** Left: Surface activation maps of saliency-sensitive response ($SS_{fMRI}$ in percent signal change) in different cortical depths of V1 in a representative participant (same participant as in B). Dashed circles indicate the location of foreground on the cortical surface. Right: Saliency maps (averaged across all participants) in image space. The data underlying this Figure can be found in *data/main_data.xlsx* at https://www.scidb.cn/doi/10.1101/2025.04.10.648136.

(Fig 3C, left). The saliency maps in the image space were reconstructed using a population receptive field (pRF) model, shown in the right panels of Fig 3C. Saliency signals were strongest in the superficial layer of V1 at the location of fore-ground bars. These findings are consistent with the V1 hypothesis in Fig 1, in which the saliency signals initially emerge by horizontal interactions in the superficial layers of V1 and subsequently feed forward to the middle layers of IPS.

### Behavioral sensitivity to saliency correlated with the saliency-sensitive fMRI response in the superficial layers of V1

To investigate whether the fMRI responses can predict the behavioral sensitivities to the foreground bars, we calculated Pearson's correlations between the behavioral sensitivity to saliency $SS_{behavior}$ and the saliency-sensitive fMRI response $SS_{fMRI}$. For each brain region, the cortical depth with the strongest saliency-sensitive fMRI response was selected. $SS_{behavior}$ showed a significant correlation with $SS_{fMRI}$ in the superficial layers of V1 (Fig 4A, $r = 0.542$, $p = 0.014$), but not with those in the superficial layers of V2 ($r = -0.218$, $p = 0.356$) or the middle layers of IPS ($r = -0.203$, $p = 0.390$) (see correlations with all layers in S11 Fig). The correlation in V1 remained significant after family-wise error (FWE) correction by permutation test (Fig 4B). The correlation results supported that the saliency-sensitive fMRI response in the superficial layers of V1 can predict the behavioral sensitivity to saliency. Although the sample size ($N = 20$) of this study is not suffi-cient for an exploratory whole-brain correlation analysis with behavioral performance, as highlighted by [51], it provides enough power for hypothesis-driven analyses focused on a limited number of brain regions [18].

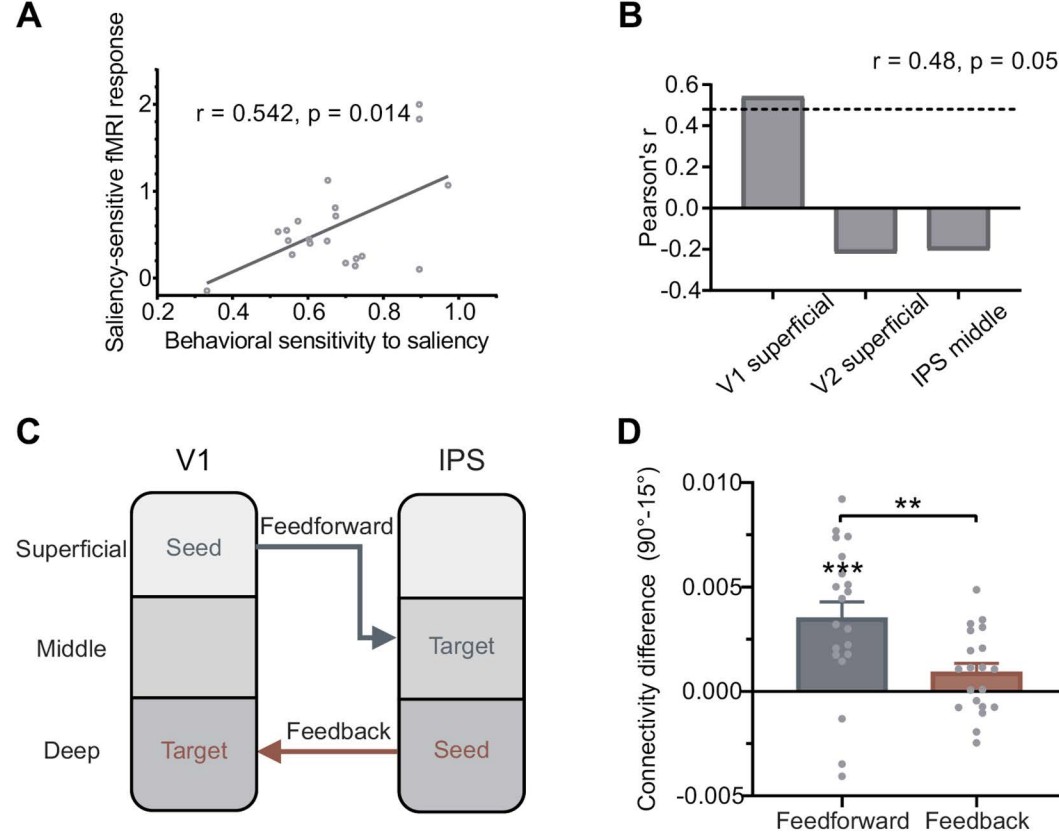

**Fig 4. Behavioral-fMRI correlations and functional connectivity. (A)** Behavioral sensitivity to saliency $SS_{behavior}$, i.e., the difference in luminance contrast sensitivity to detect the two different foregrounds, is correlated with fMRI saliency-sensitive response $SS_{fMRI}$ in the superficial layers of V1, giving a significant Pearson's correlation coefficient $r$. Each circle represents one participant. **(B)** Pearson's correlation coefficients $r$ in the superficial layers of V1 and V2, and in the middle layer of IPS. The dashed line indicates the significance threshold after family-wise error (FWE) correction by a permutation test. **(C)** The feedforward and feedback pathways of saliency signals between V1 and IPS, assessed by the layer-specific generalized psychophysiological interaction (gPPI) analysis of functional connectivity. **(D)** Functional connectivity difference (90° vs. 15°) of the two pathways in **C**. Each gray dot represents a single participant. Error bars represent the standard errors of the mean across participants. **, *** indicates $p < 0.01$, $p < 0.001$. The data underlying this Figure can be found in *data/main_data.xlsx* at https://www.scidb.cn/doi/10.1101/2025.04.10.648136.

## Layer-dependent effective connectivity revealed saliency transmission from V1 to IPS

A generalized psychophysiological interaction (gPPI) analysis was conducted to examine layer-dependent effective connectivity of saliency signals between V1 and IPS. For the feedforward connectivity $\beta_{FF}(\theta)$, the superficial layer of V1 was used as the seed ROI, and the middle layer of IPS served as the target ROI. For the feedback connectivity $\beta_{FB}(\theta)$, the seed ROI was the deep layer of IPS, and the target ROI was the deep layer of V1 (Fig 4C). The gPPI results revealed a significant connectivity difference $\beta_{FF}(90°) - \beta_{FF}(15°)$ between the 90° and 15° foreground conditions for the feedforward connection from V1 superficial to IPS middle ($t_{19} = 4.348$, $p < 0.001$, Holm corrected). In contrast, no significant difference $\beta_{FB}(90°) - \beta_{FB}(15°)$ was observed for the feedback connection from IPS deep to V1 deep ($t_{19} = 2.041$, $p = 0.055$) (Fig 4D, see S12A Fig for the original connectivity $\beta_{FF}(\theta)$ and $\beta_{FB}(\theta)$ for each $\theta$). A paired $t$ test showed that the feedforward connectivity difference $\beta_{FF}(90°) - \beta_{FF}(15°)$ was significantly stronger than the feedback connectivity difference $\beta_{FB}(90°) - \beta_{FB}(15°)$ ($t_{19} = 3.173$, $p = 0.005$). Due to the low SNR of the VASO signal, here we used the BOLD signals for the gPPI analysis. The gPPI results using VASO and BOLD signals were similar (S12B Fig), both showing a

significant feedforward connection from V1 to IPS. The connectivity differences across all cortical depths are shown in S12B Fig, indicating that the middle layer of IPS received significant feedforward input from V1. The layer-dependent connectivity results further support the hypothesis that the saliency signal propagates along the feedforward pathway from V1 to IPS.

### Fixation stability was not affected by the presence of foreground bars

Since participants were performing a fixation task in the fMRI experiment, the fixation stability was unlikely to be affected by the presence of the task-irrelevant foreground. To confirm that participants could maintain fixation, we monitored the eye movements of 12 participants in a control behavioral experiment using the same stimuli and fixation task as in the fMRI experiment. Fixation stability was assessed using the bivariate contour ellipse area (BCEA) of 95% fixations [52] (S3A Fig). BCEA results showed no significant difference among the three $\theta$ conditions ($F_{2, 22} = 0.291$, $p = 0.750$, $BF_{10} = 0.227$, moderate evidence for $H_0$: no difference across $\theta$ conditions) (S3B Fig). The distributions of gaze positions below the center of the fixation cross also showed no significant difference among the three $\theta$ conditions ($F_{2, 22} = 2.197$, $p = 0.135$, $BF_{10} = 0.736$, weak evidence for $H_0$) (S3C Fig), suggesting that participants maintained good fixation when the foreground bars were presented in the lower visual field. Therefore, our findings were unlikely to be due to unstable fixation.

## Discussion

Using 7T VASO-CBV fMRI at submillimeter resolution, we investigated layer-dependent responses to salient visual input in the early visual cortex and the parietal cortex of human participants, and the functional connectivity between these brain regions to elucidate how saliency signals are propagated. Saliency-sensitive responses were strongest in the superficial layers of V1, and in the middle layers of IPS. Meanwhile, behavioral sensitivity to saliency correlated significantly with the fMRI saliency-sensitive responses in the superficial layers of V1, but not with those in the middle layers of IPS. Furthermore, layer-dependent effective connectivity revealed significant feedforward transmission of saliency signals from V1 to IPS. These findings support the hypothesis that a bottom-up saliency map is created by contextual influences mediated by horizontal connections, which are most abundant in the superficial layers of V1, and that the saliency signals are subsequently propagated to other brain regions, such as IPS and SC, to guide attention and eye movements. Finally, the saliency signal observed in the superficial layers of V1 is unlikely to originate from the colliculus–pulvinar pathway, as this would have resulted in a similar laminar pattern in IPS (Fig 1).

While the V1SH is supported by substantial behavioral evidence with features dimensions in orientation, color, motion directions, eye of origin, and their combinations [12,53–57], neuroscience evidence remain limited and controversial [15,17,18,22]. With a large brain coverage and a submillimeter resolution, the laminar fMRI results in our study provide important mesoscale evidence for the cortical microcircuitry underlying the saliency computations for bottom-up attentional attraction in humans. The laminar response patterns from multiple brain regions (Fig 3A and 3B) and the layer-dependent connectivity results (Fig 4C and 4D) are consistent with a canonical feedforward pathway from V1 to IPS [45,46]. Since the intracortical horizontal connections in V1 are most abundant in the superficial layers [58–60], and since contextual influences mediated by these connections are key mechanisms underlying the saliency computation in the V1SH [8], our finding that the strongest saliency-related signal is in the superficial layers of V1 strengthens the evidence for V1 as the source of the saliency signals.

While the previous studies cannot rule out the SC as a potential source for saliency computation, our findings provide evidence against this possibility. If the saliency map was generated in the SC, the saliency signals should be stronger in the middle layers than in the superficial layers of V1 (Fig 1), since the thalamic input mediated from the SC via the tecto-thalamo-cortical pathway is strongest in the middle layers of V1 [48,49]. Our laminar results (Fig 3) are not consistent with this prediction. Moreover, our findings are aligned with recent evidence that visual responses in the SC of

awake primates depend critically on signals routed through the LGN and V1 [61]: deactivation of the LGN abolished SC responses to visual stimuli. It is therefore more likely that the saliency signals are generated in V1 and then projected to the superficial layers of the SC to guide attention and eye movements [19,62].

In contrast to the results reported by [18], we observed significant saliency-sensitive signals in IPS (Fig 3A and 3B). This discrepancy may be attributed to the higher spatial resolution of our 7T fMRI data, which allowed better isolation of fMRI responses to the foreground from those to the background bars. Another possible reason is the visibility of the texture stimuli, which were made invisible by backward masking in Zhang and colleagues [18]. By making the stimuli invisible, the previous study largely eliminated conscious awareness-related feedback from a high-level cortex such as the IPS, while the saliency signals (as verified by their behavioral cueing effects) were still observed in V1. However, as a consequence, the feedforward propagation of the saliency signals from V1 to IPS cannot be illustrated due to the lack of the saliency signals in the high-level region. Overcoming the limitations of the previous study, our study represents important advances in demonstrating the neural origin and recipients of saliency signals. We reveal the V1 origin and the feedforward propagation of saliency signals from V1 to IPS, by allowing the manifestation of the saliency signals in the parietal cortex and by analyzing layer-specific response and connectivity. Furthermore, the laminar response pattern arising from horizontal contextual influences in the superficial layers of V1 provides critical support for the proposed mechanisms of saliency computation in the V1SH.

Considering the slow acquisition speed of the VASO sequence, and to minimize top–down anticipatory effects, we employed a block design in which a salient foreground region was presented in the lower-left or lower-right visual field in a random order and at random intervals within a 30-s block. In principle, with such a design, we cannot exclude the possibility that the responses to ipsilateral foreground stimuli were also in the ROI-averaged response of IPS. However, our IPS ROIs were primarily located in the posterior IPS (S13 Fig), which exhibits strong lateralization and clear retinotopic organization [63]. Therefore, the saliency-sensitive responses in the IPS should be predominantly driven by the contralateral foreground. Importantly, our conclusion depends on the laminar pattern rather than the retinotopic specificity of saliency signals in the IPS. Nevertheless, an event-related design would allow for measuring fMRI responses to individual foreground events. Achieving this will require further development of high-resolution fMRI methods with faster acquisition speed.

Although this study only investigated the neural mechanisms of saliency based on orientation contrast, previous behavioral studies suggest that V1 also computes saliency from other low-level features such as color, motion direction, eye of origin, and their combinations [12,53,55,57,64]. Thus, our findings are likely to generalize to saliency by contrast in all the basic feature dimensions. Although saliency-related signals have also been observed in higher brain regions [65–67], it remains unclear whether these signals are inherited from V1 or computed within these brain regions themselves. Meanwhile, saliency signals computed by V1 should be the fastest for rapid deployment of bottom-up attentional shifts. To make the deployment rapid, the direct anatomical projection from V1 to SC should be favored compared to indirect projections from V1 to SC via higher cortical areas such as the parietal and frontal cortices. Meanwhile, when bottom-up saliency is combined with top–down factors to guide attention [2], in cooperation or in competition, the feedforward connectivity such as the ones investigated in this study, should play important roles.

In summary, mesoscale evidence at the level of cortical layers resolves a longstanding controversy regarding the neural origin of saliency signals. Our results demonstrate that visual saliency is first generated by intracortical mechanisms in the superficial layers of V1 and subsequently propagated to other attentional control regions. These findings provide direct support for V1SH, and reinforce the motivation for a new and evidence-backed vision framework [68,69], which hypothesized that the attentional bottleneck starts from V1's output to downstream areas. These findings not only shed light on an early locus of neural and computational mechanisms of visual attentional selection, but also provide valuable guidance for the development of biologically inspired artificial visual systems.

## Methods and materials

### Ethics statement

Experimental protocols were approved by the Institutional Review Board of the Institute of Biophysics, Chinese Academy of Sciences (No. 2012-IBP-011). All participants gave written informed consent before the experiments.

### Participants

Twenty healthy volunteers (9 females, aged 22–42 years) participated in the psychophysical and fMRI experiments. The sample size was determined based on the large effect (Cohen's f > 0.4) of orientation contrast on V1 responses in a previous study [18], as calculated by G*Power [70]. A sample size of 20 participants was sufficient to achieve 90% statistical power for detecting a large effect in a within-subjects ANOVA. Twelve volunteers (5 females, 22–30 years of age) participated in the control movement with eye movement recordings, 6 of them also participated in the fMRI experiment. All participants had normal or corrected-to-normal vision.

### Stimuli and procedures

Visual stimuli were generated in MATLAB (Mathworks) with Psychophysics Toolbox extension [71,72] on a Windows 10 operating system computer. Stimuli and procedures were illustrated in Fig 2A and 2C. Each texture stimulus (12.5 by 6.5 degrees of visual angle) consisted of 17 × 9 bars of 0.5° × 0.05° in visual angle presented in the lower visual field on a gray background, with 0.75° spacing between the bars. In the foreground present conditions, all bars were identical except for a foreground of 2 × 2 bars titled either 90° or 15° (orientation contrast $\theta$) relative to the background bars. The foreground bars were located at 4.3° of eccentricity either to the lower-left or lower-right of the fixation. In the uniform texture condition, all bars were uniformly oriented. The orientation of background bars was randomly chosen from 0° to 180°.

**Psychophysical experiment.** Visual stimuli were presented on a Cambridge Research System Display++ LCD monitor (32 inches, 1,920 × 1,080 pixels, 120-Hz refresh rate) at a viewing distance of 100 cm. Subjects' heads were stabilized using a chin rest.

A bright fixation cross (73 cd/m²) was shown at the center of the screen. Participants were informed to keep fixation throughout the experiment. A trial began after a button press. After 1 s of fixation, the texture stimulus was presented for 200 ms, followed by another 1 s of fixation for the participant to respond (Fig 2A) before the next trial began. Participants were instructed to make 2-alternatives-forced-choice response by pressing one of two buttons to indicate whether the foreground appeared in the lower left or lower right visual field. Bar luminance $L_{bar}$ (ranged from 43.8 to 78.9 cd/m²) was adjusted by a 3-down-1-up staircase procedure to determine the participant's contrast sensitivity for detecting the foreground. The luminance contrast of the bars from the uniform gray background (background luminance $L_0$ = 43.8 cd/m²) was defined by the Michaelson contrast $C = \frac{L_{bar} - L_0}{L_{bar} + L_0}$. The initial contrast of the staircase procedure was 0.2863 and changed by 15% in each step. Two blocks of trials (120 trials each) were collected for each orientation contrast condition. Each block consisted of two independent staircases (60 trials each) that were randomly intermixed across trials. Thus, a total of four 60-trial staircases were collected for each condition. A psychometric curve was fitted by a Weibull function relating the performance in each condition to the contrast of the bars. The contrast detection threshold $C_{threshold}$ was defined as the contrast level at 80% accuracy, and then the contrast sensitivity was defined as $S = 1/C_{threshold}$.

**FMRI experiment.** Stimuli were presented with an MRI safe projector (1,024 × 768 pixels @ 60 Hz) on a translucent screen placed behind the head coil. Participants viewed the stimuli through a mirror mounted inside the head coil at a viewing distance of 69 cm from the screen. Visual stimuli were identical as those in the psychophysical experiment except that the luminance of the bars and the uniform background were $L_{bar}$ = 87.5 cd/m² and $L_0$ = 0.4 cd/m² (max and min luminance of the projector, a high contrast texture stimulus was used to maximize the SNR of fMRI signals to the visual stimuli), respectively. The purpose of using a dark background was to enhance activations to the texture stimuli. Texture stimuli with different

orientation contrast with the background bars ($\theta = 90°$, $15°$, or $0°$) were presented in separated 30-s blocks, interleaved with 15-s fixation periods (Fig 2C). Each fMRI run lasting 270 s, with 6 blocks of stimuli per run (2 blocks for each $\theta$). In each block, there were 60 texture stimuli, each lasting 200 ms followed by an interval which was randomly 200, 300, or 400 ms. The foreground bars were randomly presented in the lower-left or lower-right visual field. Subjects were instructed to maintain fixation and count the number of subtle luminance changes (about 25% dimming) in the fixation cross. The number of luminance reductions varied across runs and was reported at the end of each run, followed by feedback. A total of 9 runs were collected for each participant. The order of stimulus conditions was balanced across runs and participants.

Two localizer runs were used for each participant to localize the retinotopic ROIs for the foreground bars. Each run consisted of eight 30-s stimulus blocks, interleaved with 15-s fixation intervals (S4 Fig). In each stimulus block, colored naturalistic stimuli (same size as the foreground in the main experiment) were presented at 4 images per second (116.7 ms presentation interleaved with 133.3 ms fixation), at one of the foreground locations. Participants were instructed to maintain fixation and to count fixation changes as in the main experiment.

**Eye tracking experiment.** Eye movements were recorded using an SR Research Eyelink 1000 Plus eye tracker. The centroids of the left eye's pupil were sampled at 1,000 Hz. Visual stimuli were presented on a 27 inches LCD monitor at $1,920 \times 1,080$ resolution and 60 Hz refresh rate, at a viewing distance of 100 cm. Participants stabilized their heads using a chin rest. The texture stimuli and fixation task were identical to those employed in the fMRI experiment. Eye gaze positions were preprocessed in the following steps: eye blinks removal (200 ms before and after the eye blinks) with linear interpolation of removed data, linear detrend, baseline correction per stimulus block. A 95% BCEA was used to quantify the distribution of fixation samples [52].

## MRI data acquisition

MRI data were collected on a 7T scanner (Siemens MAGNETOM) with a 32-channel receive single-channel transmit head coil (Nova Medical) in the Beijing MRI Center for Brain Research. The gradient coil had a maximum amplitude of 70 mT/m, 200 μs minimum gradient rise time, and 200 T/m/s maximum slew rate. Participants used a bite bar to reduce head motion. Functional data of blood nulled and not-nulled images were acquired using a SS-SI-VASO sequence with 3D-EPI readout [50,73] (0.82-mm isotropic voxels, 26 oblique-coronal slices, FOV = $133 \times 177$ mm$^2$, paired-TR = 5,020 ms, TE = 25 ms, TI1 = 1,744 ms, FA = 26°, bandwidth = 1,064 Hz/pixel, Partial Fourier = 6/8 with 8 POCS iterations, GRAPPA = 3 with FLASH reference). The 3D slab of fMRI acquisition was placed in oblique-coronal orientation (S1C Fig). Images with reversed phase and read directions were acquired for susceptibility distortion correction. T1-weighted anatomical images were acquired using an MP2RAGE sequence [74] (0.7-mm isotropic voxels, FOV = $224 \times 224$ mm$^2$, 256 sagittal slices, TE = 3.05 ms, TR = 4,000 ms, TI1 = 750 ms, FA = 4°, TI2 = 2,500 ms, FA = 5°, bandwidth = 240 Hz/pixel, phase and slice partial Fourier = 7/8, GRAPPA = 3).

## MRI data analysis

**Preprocessing of functional data.** The preprocessing of fMRI data was performed using AFNI [75], LAYNII [76] and custom Python code. The following steps were applied for blood-nulled and not-nulled volumes separately: EPI image distortion correction with blip-up/down nonlinear warping method, rigid-body correction of head motion, spatially up-sampled by a factor of 2 (3dResample in AFNI) [77]. Since the blood-nulled and not-nulled volumes were acquired alternately, the 3D timeseries were up-sampled by a factor of 2 (3dUpsample in AFNI) and shifted by TR/2 for BOLD correction. The VASO or CBV-weighted volume was calculated by dividing the nulled by not-nulled volumes (LN_BOCO in LAYNII). To minimize image blur, all spatial transformations were combined and applied to the functional images in one interpolation (3dAllineate in AFNI with sinc method). After per run scaling as percent signal change, a general linear model (GLM) with a canonical HRF (BLOCK4 in AFNI) was used to estimate the VASO and BOLD signal change from baseline. The fMRI signals in all cortical depths were fitted equally well by convolving the stimulus regressor with the canonical HRF

(S14 Fig). Thus, the GLM analysis should not introduce bias across cortical depth. Head motions and low-order drifts were included as regressors of no interest in the GLM.

**Surface segmentation.** The T1-w anatomical volume was aligned to the mean EPI image after motion correction, and segmented into WM, gray matter (GM), and CSF compartments using FreeSurfer with the "high-res" option [78]. FreeSurfer's segmentation results were visually inspected and manually edited to remove dura mater and sinus, etc., ensuring accurate surface segmentation (S1A Fig). High-density surface mesh was generated by a factor of 4 to better align with the up-sampled volume grid [79]. The cortical depth profiles were constructed using the equi-volume model [80]. Two equi-volume intermediate surfaces between the WM and pial surfaces were generated using mripy ([https://github.com/herrlich10/mripy](https://github.com/herrlich10/mripy)), dividing the GM into three equi-volume layer compartments (S1A Fig). For each voxel, the layer weight (volume percentage among WM, CSF, and the three-layer compartments) were calculated and subsequently used in a spatial regression approach to unmix layer activity [81]. Specifically, We applied a general linear model (GLM) to estimate the response associated with each cortical layer: $Y = X \times B + U$, where $Y$ is an $n \times 1$ vector containing voxel responses within the ROI ($n$ denotes the total number of voxels), $X$ is an $n \times 5$ layer weight matrix representing the proportion of each voxel's volume assigned to the five compartments (three cortical layers, WM, and CSF), and $B$ is a $5 \times 1$ vector of the estimated layer-specific responses. The residual term $U$ is an $n \times 1$ vector capturing deviations from the least-squares fit. This unmixing approach helps to minimize correlations in activity across different cortical layers.

**ROI definition.** Anatomical ROIs of the early visual areas (V1–V2) were defined on the cortical surface by a 7T retinotopic atlas of the Human Connectome Project using Neuropythy tools [82,83]. The anatomical ROI for IPS was taken from the Wang15 atlas [84]. Clusters of voxels with significant positive activation in the localizer ($p < 0.05$ uncorrected, cluster size ≥ 120 upsampled voxels) within the anatomical boundary of IPS were selected as the foreground ROIs (Fig 2D). Three participants (five hemispheres) did not show significant clusters at the threshold. In these cases, we slightly relaxed the cluster-defining threshold until two clusters could be identified for each hemisphere. The foreground ROIs in IPS consisted of 2–3 clusters per hemisphere ($2.625 \pm 0.807$ on average), mainly located in the posterior IPS (S13B Fig). Accordingly, surround ROIs in V1 and V2 were acquired by subtracting the foreground ROI from vertices with significant activation to the texture stimuli ($p < 0.05$, uncorrected). Surface ROIs were then projected to the EPI volume to select voxels in a column-wise manner. Finally, the numbers of (up-sampled) voxels corresponding to different layers of the V1, V2, and IPS were: V1 deep: $352.6 \pm 41.0$ (Mean ± SEM across participants); V1 middle: $327.0 \pm 39.6$; V1 superficial: $294.2 \pm 36.2$; V2 deep: $371.6 \pm 49.3$; V2 middle: $341.8 \pm 44.7$; V2 superficial: $310.7 \pm 41.4$; IPS deep: $1816.7 \pm 460.9$; IPS middle: $1748.5 \pm 449.1$; IPS superficial: $1630.0 \pm 435.9$.

**Visual field map reconstruction.** The visual field map of saliency-related signals was reconstructed for each cortical depth using the saliency-dependent ($\theta_{90} - \theta_{15}$) CBV responses and the pRF maps from the Benson atlas [83,85]. To account for individual differences in the pRF maps, the pRF locations were linearly transformed by aligning the peak of localizer activation to the foreground location. Specifically, angle and eccentricity parameters ($A_m$ and $E_m$) were obtained for the cortical vertex showing the maximum localizer activation, separately for each hemisphere. The corrected pRF parameters were defined as:

$$A_c = \begin{cases} \frac{A}{A_m} \times 127.88°, & 0° \leq A \leq A_m \\ 180° - \frac{180° - A}{180° - A_m} \times (180° - 127.88°), & A_m < A \leq 180° \end{cases}, \quad E_c = \frac{E}{E_m} \times 4.28°.$$

Here, the polar angle of 127.88° and eccentricity of 4.28° correspond to the screen location of foreground bars. $A$ (0°–180° within each hemisphere) and $E$ are the original pRF parameters provided by the Benson atlas [83].

The visual field map was reconstructed by multiplying the each node's CBV response with its corrected pRF map and then summed across all nodes [86]. Finally, the reconstructed map was normalized (divided) by the maximum response across all cortical depths, and then averaged across participants.

**Layer-dependent effective connectivity.** A gPPI method was used to investigate the effective connectivity across cortical depths in V1 and IPS [87]. CBV timeseries of the foreground ROI were averaged within each cortical depth of V1 and IPS, and then normalized by z-score. Due to the uncertainty about CBV HRF, the PPI term was calculated by the dot product of the stimulus regressor (boxcar function, shifted by one TR due to the HRF delay) and the seed timecourse without deconvolution [88,89]. The GLM includes the seed region time course, stimulus regressors, PPI terms of the three $\theta$ conditions, along with baseline regressors and head motion parameters. Saliency-dependent functional connectivity was defined as the difference between the PPI beta weights of the $\theta_{90}$ and $\theta_{15}$ conditions. To ensure retinotopic correspondence between V1 and IPS activity, gPPI analysis was performed within each hemisphere, and then results from the two hemispheres were averaged.

## Statistical analysis

Two-way rm ANOVA with cortical depths (deep, middle, and superficial) and $\theta$ (90°, 15°, and 0°) as within-subject factors was performed on the ROI-averaged CBV responses. All data met the sphericity assumption of ANOVA. In cases where the sphericity assumption was violated, $p$ values were adjusted using the Greenhouse-Geisser correction. One-way rm ANOVA was performed to evaluate the difference in saliency-related signals across cortical depths, followed paired $t$ tests if there was a significant difference. No FWE correction is needed for paired $t$ tests followed by a significant one-way ANOVA with three levels [90]. To examine whether the laminar profiles differed across cortical regions, a two-way rms ANOVA was conducted on the saliency-related signals, with cortical depth and ROI (V1, V2, IPS) as within-subject factors.

Pearson's r was calculated to assess the correlation between the fMRI saliency sensitivity $SS_{fMRI}$ and the behavioral saliency sensitivity $SS_{behavior}$. Data were assessed for multivariate normality using Shapiro–Wilk test (all $p > 0.05$). A permutation test was performed to correct for the FWE across ROIs. In each permutation, Pearson's r was calculated after shuffling the correspondence between fMRI signal and perceptual sensitivity across participants. The largest r value was taken and the permutation was repeated to generate a null distribution, from which the FWE was derived for the original r values.

## Supporting information

**S1 Fig. Cortical depth segmentation and scanning region. (A)** Normalized cortical depth map overlayed on the T1w anatomical image in a representative participant. The equi-volume depth at 0 and 1 correspond to the WM and Pial surfaces, respectively. **(B)** VASO activations (90° + 15° + 0°, $p < 0.001$ uncorrected) overlayed on the mean VASO image. Green and yellow lines indicate the WM surface, while the blue and pink lines denote the pial surface. **(C)** The green box indicates the 3D slab of VASO fMRI acquisition.
(TIF)

**S2 Fig. Subjects' performance of the fMRI saliency experiment task.** Accuracy represented the proportion of the runs that the participants gave the correct numbers of fixation changes. Deviation represented the difference between the giving numbers and the real numbers divided by the real numbers. The data underlying this Figure can be found in *data/ S2_data.xlsx* at https://www.scidb.cn/doi/10.1101/2025.04.10.648136.
(TIF)

**S3 Fig. Results of eye tracking experiment. (A)** The group-averaged heat maps of gaze density in the entire session. **(B)** The bivariate contour ellipse area (BCEA) of fixation distribution showed no significant difference across $\theta$ conditions ($F2,22 = 0.291$, $p = 0.750$, BF10 = 0.227, moderate evidence for H0: no difference across $\theta$ conditions). Error bars represent the standard deviation of the means. **(C)** The percentages of the gaze positions (gaze density) below the center of the fixation point showed no significant difference across $\theta$ conditions ($F2,22 = 2.197$, $p = 0.135$, BF10 = 0.736, weak evidence

for H0). The dash lines indicate the region of the gaze positions selected for analysis. The data underlying this Figure can be found in *data/S3_data.xlsx* at https://www.scidb.cn/doi/10.1101/2025.04.10.648136.
(TIF)

**S4 Fig. Stimulus and procedure of the localizer runs.** Naturalistic stimuli were presented at four images per second in the lower-left or lower-right quadrants in separate stimulus blocks, interleaved with 15-s fixation periods. The size and location of localizer stimuli matched the foreground region in the main experiment.
(TIF)

**S5 Fig. Normalized BOLD responses in the foreground ROIs of V1, V2, and IPS.** Top panel: BOLD response in different depths of V1, V2, and IPS in 90°, 15°, and 0° orientation contrast conditions; Bottom panel: Calculated from top panel, the response difference between 90° and 15° foregrounds. Error bars represent the standard deviation of the mean. *, ** and *** indicate $p < 0.05$, $p < 0.01$, $p < 0.001$. The data underlying this Figure can be found in *data/S5_data.xlsx* at https://www.scidb.cn/doi/10.1101/2025.04.10.648136.
(TIF)

**S6 Fig. Laminar profile of CBV responses extended to into WM and CSF.** Top panel: CBV responses ($S_{fMRI}$ in percent signal change) to the orientation foregrounds in V1, V2, and IPS. Bottom panel: CBV response difference between the 90° and 15° foreground conditions ($SS_{fMRI} = S_{fMRI}(90°) - S_{fMRI}(15°)$). Error bars represent SEM. WM: white matter, GM: gray matter, CSF: cerebrospinal fluid.
(TIF)

**S7 Fig. Normalized CBV responses in the background ROIs of V1 and V2.** Upper: A significant effect of orientation contrast ($\theta = 90°$, 15°, and 0°) was found in V1 ($F_{2,38} = 3.700$, $p = 0.034$) and a similar trend in V2, suggesting weaker background activity in the 90° and 15° conditions compared to the 0° or the uniform texture condition. No significant difference was found between the two $\theta$ conditions ($F_{1,19} = 0.365$, $p = 0.553$, BF10 = $5.601 \times 10^{-11}$). Lower: The suppression effect was calculated as the response difference between the mean of 90° and 15° conditions and the 0° condition (($90° + 15°)/2 - 0°$). A significant suppression effect was found only in the superficial depth of V1 ($t_{19} = -2.877$, $p = 0.023$, Holm corrected across cortical depths). These results suggest a suppression effect of background activity in the superficial layers of V1, independent with the orientation contrast between the foreground and the background bars. Each gray dot represents one participant. Error bars indicate SEM. * indicates $p < 0.05$. D, M, S indicate deep, middle, and superficial depth, respectively. The data underlying this Figure can be found in *data/S7_data.xlsx* at https://www.scidb.cn/doi/10.1101/2025.04.10.648136.
(TIF)

**S8 Fig. Unnormalized (or original) CBV responses in the foreground ROIs.** Conventions are identical as in Fig 3A. The data underlying this Figure can be found in *data/S8_data.xlsx* at https://www.scidb.cn/doi/10.1101/2025.04.10.648136.
(TIF)

**S9 Fig. Normalized CBV response difference in the foreground ROIs between 90° and 0° orientation contrast conditions.** Similar laminar profile with Fig 3A bottom panel. Error bars represent the standard error of the mean. * $p < 0.05$, + $p < 0.1$. The data underlying this Figure can be found in *data/S9_data.xlsx* at https://www.scidb.cn/doi/10.1101/2025.04.10.648136.
(TIF)

**S10 Fig. SNRs across cortical depths and the depth-dependent saliency signals normalized by SNR. (A)** The tSNR and CNR of CBV response across cortical depth in the foreground ROIs. tSNR: temporal signal-to-noise ratio, mean

signal divided by the standard deviation. CNR: contrast-to-noise ratio, signal difference between stimulus and fixation periods divided by the standard deviation. **(B)** Saliency-sensitive responses normalized by tSNR. **(C)** Saliency-sensitive responses normalized by CNR. Error bars represent the standard error of the mean. *, **, and *** indicate $p < 0.05$, $p < 0.01$, and $p < 0.001$, respectively. The data underlying this Figure can be found in *data/S10_data.xlsx* at https://www.scidb.cn/doi/10.1101/2025.04.10.648136.
(TIF)

**S11 Fig. Pearson's correlation between $SS_{behavior}$ and $SS_{fMRI}$ across three cortical depths in V1, V2, and IPS.** Each circle represents one participant.
(TIF)

**S12 Fig. Original connectivity beta values and connectivity matrix. (A)** Beta values of gPPI terms for the 90° and 15° conditions. Each gray dot represents one participant. Error bars represent the standard errors of the mean across participants. One sample *t* tests indicate significant positive connectivity in 90° of the feedforward pathways, 90° and 15° of the feedback pathways (feedforward 90°: $t_{19} = 5.853$, $p < 0.001$; feedforward 15°: $t_{19} = 1.253$, $p = 0.225$; feedback 90°: $t_{19} = 4.350$, $p < 0.001$; feedback 15°: $t_{19} = 3.734$, $p = 0.001$). Paired *t* tests indicate significant connectivity difference only in the feedforward pathway (feedforward 90° vs. 15°: $t_{19} = 4.384$, $p < 0.001$; feedback 90° vs. 15°: $t_{19} = 2.041$, $p = 0.055$). A two-way rm ANOVA shows a significant interaction between pathways (feedforward and feedback) and $\theta$ conditions (90° and 15°) ($F_{1,19} = 10.067$, $p = 0.005$). **(B)** The gPPI connectivity matrix across three cortical depths in V1 and IPS. Columns and rows correspond to the seed and target ROIs, respectively. The color scale indicates the beta difference of interaction terms (beta(90°) − beta(15°)). * $p < 0.05$, ** $p < 0.01$, *** $p < 0.001$, uncorrected. The data underlying this Figure can be found in *data/S12_data.xlsx* at https://www.scidb.cn/doi/10.1101/2025.04.10.648136.
(TIF)

**S13 Fig. The lateralization of localizer activations in IPS and the ROI distributions in IPS subregions. (A)** Localizer activations to the left- or right-side stimuli in the IPS of a representative participant. Dashed lines mark the borders of the IPS. **(B)** Group-averaged ROI volume in IPS subregions. Error bars represent SEM across participants. Each gray dot represents data from one participant. The data underlying this Figure can be found in *data/S13_data.xlsx* at https://www.scidb.cn/doi/10.1101/2025.04.10.648136.
(TIF)

**S14 Fig. VASO and BOLD response timecourses in deep, middle, and superficial ROIs of the foreground in V1, and the fitted response with GLM using a canonical HRF (BLOCK4 in AFNI).** Error bars represent SEM across participants.
(TIF)

## Author contributions

**Conceptualization:** Peng Zhang.

**Data curation:** Chen Liu, Chengwen Liu, Peng Zhang.

**Formal analysis:** Chen Liu, Chengwen Liu, Peng Zhang.

**Funding acquisition:** Peng Zhang.

**Investigation:** Chen Liu, Chengwen Liu, Peng Zhang.

**Methodology:** Chen Liu, Chengwen Liu, Laurentius Huber, Peng Zhang.

**Project administration:** Peng Zhang.

**Resources:** Peng Zhang.

**Software:** Chen Liu, Chengwen Liu, Laurentius Huber, Peng Zhang.

**Supervision:** Peng Zhang.

**Validation:** Peng Zhang.

**Visualization:** Chen Liu, Peng Zhang.

**Writing – original draft:** Chen Liu, Peng Zhang.

**Writing – review & editing:** Chengwen Liu, Li Zhaoping, Peng Zhang.

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
