## [Editor Report · Decision Letter 0]

7 Apr 2025

Dear Dr Zhang, 

Thank you for submitting your manuscript entitled "V1 superficial layers create a saliency map that feeds forward to the parietal cortex for attentional guidance" for consideration as a Short Reports by PLOS Biology.

Your manuscript has now been evaluated by the PLOS Biology editorial staff, as well as by an academic editor with relevant expertise, and I am writing to let you know that we would like to send your submission out for external peer review.

Once your full submission is complete, your paper will undergo a series of checks in preparation for peer review. After your manuscript has passed the checks it will be sent out for review. To provide the metadata for your submission, please Login to Editorial Manager (https://www.editorialmanager.com/pbiology) within two working days, i.e. by Apr 09 2025 11:59PM.

Kind regards,

Taylor

Taylor Hart, PhD, 

Associate Editor

PLOS Biology

thart@plos.org

---

## [Decision Letter · Decision Letter 1]

30 May 2025

Dear Dr Zhang,

Thank you for your patience while your manuscript "V1 superficial layers create a saliency map that feeds forward to the parietal cortex for attentional guidance" was peer-reviewed at PLOS Biology. It has now been evaluated by the PLOS Biology editors, an Academic Editor with relevant expertise, and by several independent reviewers. 

In light of the reviews, which you will find at the end of this email, we would like to invite you to revise the work to thoroughly address the reviewers' reports.

As you will see, the reviewers describe the topic as interesting and the methods used as state of the art. However, they also raised a number of concerns over missing analytical, methodological, and statistical details, and insufficient explanations/justifications. In preparing your revision, you should carefully consider the points raised by the reviewers and respond to them thoroughly. We think that you should pay particular attention to Reviewer 3's concerns related to the task, task relevance, and task difficulty, and also ensure that your revised manuscript provides a better engagement with prior fMRI studies of saliency.

Given the extent of revision needed, we cannot make a decision about publication until we have seen the revised manuscript and your response to the reviewers' comments. Your revised manuscript is likely to be sent for further evaluation by all or a subset of the reviewers.

**IMPORTANT - SUBMITTING YOUR REVISION**

*Re-submission Checklist*

*Published Peer Review*

*PLOS Data Policy*

*Blot and Gel Data Policy*

Sincerely,

Taylor

Taylor Hart, PhD, 

Associate Editor

PLOS Biology

thart@plos.org

REVIEWS:

Reviewer #1: This study uses laminar fMRI to investigate whether saliency signals are generated in V1, Superior Colliculus or in IPS, as a way to arbitrate between competing hypotheses in the literature. Different laminar profiles of saliency modulation would be expected depending on the source of the saliency signal. High-field fMRI in combination with a CBV-weighted VASO technique is employed to measure activation in different cortical layers while participants are presented with low-level stimuli varying in saliency. The results show that saliency effects were strongest in superficial layers in V1 and in middle layers in IPS, in line with the V1 Saliency Hypothesis proposing that saliency is generated in V1. 

The paper is well written and uses state-of-the-art methods to investigate a clearly defined and interesting question. I was especially impressed by the clarity with which the hypotheses were formulated and investigated, although I also have some questions about that (see below). That being said, I still have a few concerns that I would like to see addressed. 

Major comments:

The hypotheses entirely rely on the assumption that the involvement of specific layers corresponds to the direction of communication (forward or backward). The authors cite the classic Felleman and van Essen (1991) study to motivate their laminar hypotheses (line 116). However, is this the only work supporting the direction of communication between the layers of different cortical cites or has other research provided convergent findings? For example, how many studies have shown that feedback connections from parietal to V1 go from deep to deep and superficial layers? To be able to interpret the results, it is crucial to indicate how solid the evidence for this assumption is. 

Relatedly, it was unclear to me whether the hypothesized laminar profiles in Fig. 1 were generated using some kind of computational model or illustrated by hand based on the descriptions in the literature? Please clarify in the text. 

I was a bit surprised to read that the analyses focused on the laminar profiles in V1, V2 and IPS while the introduction and hypotheses are focused on V1, SC and IPS. Eventually I realized that SC was not included because the effects of SC being the source would be reflected in the laminar profile of V1. However, it was still not entirely clear to me why V2 was included. It would be helpful if the authors could emphasize the relationship between SC being the source and the laminar profile in V1 and also further explain the relationship between the hypotheses and the laminar profile in V2.

I found it surprising that the saliency effect in V1, but not IPS, predicted the behaviour since we might a priori expect that responses in more downstream regions, closer to the output, are more predictive. Could the participants explain how this might work? Relatedly, between-subject correlations with such a small N are notoriously underpowered (e.g. see Marek et al., 2022 Neuron), which should be mentioned.

I was confused by the PPI analysis. The methods state that the saliency-dependent connectivity was defined as the difference between the PPI beta weights of the two conditions. However, shouldn't the PPI weight already be defined as an interaction between the time series and the difference between the two conditions (https://pubmed.ncbi.nlm.nih.gov/9344826/)? In that case, the PPI beta itself would already show the condition effect. Furthermore, in standard PPI, the analysis is executed whole-brain. How exactly is an ROI target analysis implemented here? 

Minor comments: 

For the fMRI task, was the participant adjusted luminance used or just black and white stimuli? 

Line 297 stating that the neural source of saliency has been controversial should be followed with references to this debate. 

Motivations for the study are discussed in the discussion (starting line 319) that would be better placed in the introduction. 

Reviewer #2: This study leverages 7T laminar fMRI to investigate the neural substrates of visual saliency, revealing robust saliency-related signals in superficial layers of V1 and middle layers of IPS, alongside feedforward information flow from V1 to IPS. Further, the saliency-related activity in superficial V1 layers can predict participants' behavioral performance. Overall, the research question is interesting and the manuscript is well-written. Several points require clarification to strengthen the interpretation and technical rigor.

1 the introduction posits three potential hypotheses (V1, IPS, SC) for saliency computation. While substantial empirical support is provided for the first two hypotheses, the rationale for the third hypothesis involving subcortical pathways is not very clear. Given the established role of the superior colliculus (SC) and pulvinar in attentional processes, it would be valuable to clarify whether the term "thalamus" in the introduction and Figure 1 specifically denotes the pulvinar nucleus. Furthermore, as prior work highlights the tecto-thalamo-cortical pathway (SC→pulvinar→superficial V1 layers), whether the observed superficial V1 effects might originate from this pathway need to be at least discussed?

2 minor inconsistencies in data visualization warrant verification. For instance, in Figure S3 (IPS superficial layers) and Figure S6 (V1 deeper/middle layers), the distribution of individual data points appears partially misaligned with the plotted mean values. 

3 while the VASO sequence effectively minimizes macrovascular contamination and improves spatial specificity, residual laminar signal-to-noise ratio (SNR) biases may persist. For example, Figure 3 illustrates consistently elevated superficial-layer signals across regions (green traces). Please rule out the possibility that superficial V1 effect is contaminated by the SNR differences across layers.

4 the x-axis label in Figure 4A requires clarification. Based on Figure 2B, this axis does not appear to represent behavioral sensitivity SS(behavior). A precise definition of this metric is essential for reproducibility and data interpretation.

5 While fixation stability was compared across orientation contrast conditions, a more rigorous approach might involve analyzing gaze position distributions (e.g., heatmaps or dispersion metrics) across different stimulus positions.

6 Please provide details on the number of voxels included in each laminar ROI across regions, as the activation in IPS in Figure 2D seems quite weak.

Reviewer #3: This manuscript reports the results from a psychophysical and high-resolution 7T fMRI study assaying the laminar profile of saliency modulations of neural activity evoked by task-irrelevant stimuli. The authors show evidence that superficial layers of V1 are more sensitive to changes in stimulus salience than middle/deep layers, consistent with a mechanism whereby feature contrast is computed locally in early visual cortex and transmitted to other regions, like parietal cortex. IPS was sensitive to changes in salience, but with a different laminar profile more consistent with receiving feedforward input. V1 superficial layer sensitivity to salience (difference in activation for high - low salience stimuli) correlated with individual participant sensitivity to differences in salience measured psychophysically outside the scanner (in a task where the salient location was reported, and thus task-relevant). Finally, the authors compare whether activation in V1 superficial layers drives activation in IPS middle layers, as predicted, using a gPPI analysis (version of functional connectivity), and show that connectivity is stronger when salience is greater, a result missing from feedback connections from deep IPS to deep V1 layers. The authors conclude that this establishes a key role of V1 in computing a saliency map, which is fed forward to later regions like IPS, consistent with the V1 salience hypothesis.

Overall, this is a strong study. The data is interesting, and the difference in relative activation profiles across layers between high- and low- salience conditions appears quite solid. I have some comments about the experiment design and reporting of results, as well as the degree to which conclusions from this study can be generalized to salience computations for other types of visual features. It will certainly be of interest to vision scientists, and likely to systems neuroscientists more broadly.

Major comments:

1. Throughout the manuscript, the authors discuss 'salience maps', which per my understanding index the local feature contrast at specific locations in the visual environment, typically in retinotopic coordinates. But, the actual experiment doesn't seem to address spatial sensitivity whatsoever (as far as I can tell). This is because in the fMRI study, it seems that each block contains both trials containing salient locations on the left and right side of the screen (randomly chosen for each of the 60 trials per 30 s block). This means both hemispheres' foreground-selective ROI should be stimulated, which is I suppose some degree of evidence for a 'map' (at least in V1), but this also means it's not possible to establish spatial localization of salience computations on each block. If instead left-only and right-only blocks were included, the authors could verify that results hold when the salient stimulus is presented in the contralateral visual field for each ROI (rather than anywhere on the screen), and this would allow for better testing of the functional connectivity, because the specific prediction seems to be that information about the salience of a visual field location is transmitted from V1 to IPS between corresponding parts of their respective retinotopic maps. I understand that acquiring more data is infeasible, but this limitation should be clearly considered, and in my view the impact of the limitation on the conclusions may be somewhat broad. (I'll note that it's very obvious that the signal in V1/V2 is exceedingly likely to be driven by the foreground stimulus in the contralateral visual field due to small pRFs and clear retinotopic organization. But the same isn't as clearly true for IPS, due both to large pRF size and less clear retinotopic organization. Because the results rely on this comparison, I think this issue needs to be discussed).

2. In this study, the salient stimuli are never task-relevant (inside the scanner). This is enforced using a fixation task. However, the difficulty of the fixation task is not mentioned. What was participant performance on the fixation task? I worry that if the task is too easy, the participants can be simultaneously monitoring the peripheral stimuli, and so aspects of the results could reflect covert selection/attention.

3. Additionally, related to task relevance: it may be useful to consider how computations in salience maps like that measured here are expected to change across manipulations of task relevance of salient stimuli. Clearly, the task-irrelevant stimuli can evoke location-selective fMRI responses - but what if there were a competing task, like search for color singleton? 

4. The authors interpret their findings - which I'm overall convinced by - as evidence that V1 acts as a salience map. Is it valid to interpret this as a salience map given only a single feature - for which V1 is tuned - is tested? Would the authors predict that other salient features, like color, shape, or motion direction (e.g., Burrows & Moore, 2009), are encoded in a similar manner, with V1/IPS responses scaling as a function of feature contrast and correlating with behavior? Or would these types of computations occur in other regions? Some other recent fMRI work (e.g., Thayer & Sprague, 2023; Bogler et al, 2011; Poltoratski et al, 2017) may speak to this, and is not cited in the present manuscript.

5. Statistics: at present, a core part of the central conclusion rests on the results shown in Fig. 3A - the depth profile of the saliency signal changes across ROIs. However, the critical interaction between depth and ROI is not tested (that I could see).

Minor comments:

1. Neural/behavioral correlation - right now, the 'winning' layer is used for each region, and the authors conclude a selective correlation between V1 superficial layer salience modulation and behavioral salience sensitivity. But is this result selective to superficial layers in V1, rather than middle/deep layers of V1? All layers, ROIs should be tested for completeness.

2. In the connectivity analyses, only gPPI differences between 90 and 15 deg conditions are reported. Is the gPPI positive for each of the values used to compute the difference, for each layer/ROI tested? If both values are negative, but gPPI is less negative for one condition than another, this would be more challenging to interpret as support for the authors' hypothesis

3. In the gPPI analysis, the authors report a significant connectivity difference for feedforward connections, and no significant connectivity difference for feedback - but is there a difference between these differences?

4. In Fig. 2D, it looks like there are 3 disjoint clusters identified by the localizer in IPS. Were all significant voxels included? These may aggregate across different retinotopic IPS regions (discussed in detail in the Wang et al 2015 paper describing the atlas used by the authors). Is this the case for all hemispheres/subjects? 

5. The Methods mention that the pRF-based image space visualizations shown in Fig. 3C are 'linearly transformed' to align across participants (line 527-28). Please elaborate on this process, especially because there are not individual-subject pRF maps available. (additionally, perhaps the localizer stimulus could be useful here?)

6. What were the contrast steps used for the staircases in the psychophysics experiment?

7. Fig. 3B shows the effect of salience on fMRI activation in volume space for an example participant. However, it appears that this activation is masked by the localizer-defined ROI. It may be helpful to illustrate unmasked activation profiles, including, if possible, unmasking CSF/GM to show the spatial specificity of signals attained

8. More details should be provided about how activation was mapped from voxels to layers (i.e., the demixing procedure/associated procedures should be specified in the Methods if possible)

---

## [Decision Letter · Decision Letter 2]

29 Aug 2025

Dear Dr Zhang,

Thank you for your patience while we considered your revised manuscript "V1 superficial layers create a saliency map that feeds forward to the parietal cortex for attentional guidance" for consideration as a Short Report at PLOS Biology. Your revised study has now been evaluated by the PLOS Biology editors, the Academic Editor, and the original reviewers.

In light of the reviews, which you will find at the end of this email, we are pleased to offer you the opportunity to address the remaining points from Reviewer 3 in a revision that we anticipate should not take you very long. We will then assess your revised manuscript and your response to the reviewers' comments with our Academic Editor aiming to avoid further rounds of peer-review, although we might need to consult with the reviewers, depending on the nature of the revisions.

**IMPORTANT - SUBMITTING YOUR REVISION**

*Resubmission Checklist*

*Published Peer Review*

*PLOS Data Policy*

*Blot and Gel Data Policy*

Sincerely,

Taylor

Taylor Hart, PhD, 

Associate Editor

PLOS Biology

thart@plos.org

REVIEWS:

Reviewer #1: The authors have addressed all my previous comments. This is a great paper. 

Reviewer #2 [Ke Jia]: I appreciate the authors' responses to my comments, and I find that all my concerns have been adequately addressed in the revised manuscript.

Reviewer #3: I appreciate the authors' very careful attention to my previous comments, and to comments raised by the other reviewers. In my view, the manuscript is much stronger after these edits. My only remaining comment concerns the scaling procedure used to adjust atlas-based pRF values to align with a localizer. I think this is overall a clever and efficient solution to the problem of estimating a rough mapping between single-subject activation patterns and visual field maps in the absence of subject-specific retinotopy, but I'm a bit worried about the selection of scaling procedure implemented (see below)

1. Scaling: In their detailed reply to my previously-raised point 5, the authors helpfully describe how they adjusted the pRF parameters pulled from the Benson atlas based on the participant's localizer data. However, I'm a bit confused/concerned about the scaling operation applied to polar angle - why scale linearly in polar coordinates, rather than linearly in Cartesian coordinates? If I'm understanding correctly, this type of polar angle scaling is similar to opening/closing a handheld fan - voxel pRF positions would rotationally expand/contract from 0 deg. Such a scaling approach does not account for the circular nature of the variable (i.e., a >1 scaling term will mean polar angles of 359 will 'loop back around'; a <1 scaling term would leave a portion of the visual field empty). Additionally, it seems that this approach will necessarily result in some voxels 'moving' across the horizontal hemifield boundary (e.g., a voxel with an original polar angle of 195 deg, assuming 180 deg is lower vertical meridian, could be scaled below 180 deg, which would move the pRF across the meridian). A linear scaling in Cartesian coordinates would be more appropriate. Alternatively, if the scaling factors are demonstrated to be quite small (~10% or less for polar angle adjustments), I'm not as concerned, as the impact of this choice would be minimal.

---

## [Editor Report · Decision Letter 3]

16 Sep 2025

Dear Dr Zhang,

Thank you for your patience while we considered your revised manuscript "V1 superficial layers create a saliency map that feeds forward to the parietal cortex for attentional guidance" for publication as a Short Report at PLOS Biology. This revised version of your manuscript has been evaluated by the PLOS Biology editors and the Academic Editor.

Based on our Academic Editor's assessment of your revision, we are likely to accept this manuscript for publication.

We discussed your request to change the article type from Short Report to Research Article. However, we still believe that Short Report is the best section for your paper at PLOS Biology. But please note that this makes no difference for your paper's listing on PubMed and most other databases.

Please also make sure to address the following data and other policy-related requests.

IMPORTANT: Please ensure that the next version of your paper incorporates the following changes:

------------

**Title

-- We suggest a small change to your article title: 

"The superficial layers of the primary visual cortex create a saliency map that feeds forward to the parietal cortex for attentional guidance"

**Ethics: 

-- We require a small adjustment to your Ethics / Participants statement:

1. The Materials and Methods section should start with the subheading "Ethics Statement", and include in this section the final sentence from what are currently the final two sentences from your "Participants" subsection (separating information about Ethics from information about Participants).

2. Please note that all research involving human participants must have been conducted according to the principles expressed in the Declaration of Helsinki.

https://journals.plos.org/plosbiology/s/ethical-publishing-practice

**Data:

Thank you for uploading your data to Science Data Bank. Please check that these data contain the numerical values for the following figure panels (you can also include these numerical values as a supplementary excel file):

2B

3A

4BD

S2

S3BC

S5

S7

S8

S9

S10

S12A

S13B

-- Please cite the location of the data clearly in all relevant main and supplementary Figure legends, e.g. “The data underlying this Figure can be found in S1 Data” or “The data underlying this Figure can be found in https://doi.org/10.5281/zenodo.XXXXX”

-- If you are reporting experiments where n ≤ 5, please plot each individual data point.

-- Supplementary files (e.g., excel). Please ensure that all data files are uploaded as 'Supporting Information' and are invariably referred to (in the manuscript, figure legends, and the Description field when uploading your files) using the following format verbatim: S1 Data, S2 Data, etc. Multiple panels of a single or even several figures can be included as multiple sheets in one excel file that is saved using exactly the following convention: S1_Data.xlsx (using an underscore).

**Code availability:

Thank you for making your code available on GitHub. 

Thank you for providing the underlying code in GitHub. However, because Github depositions can be readily changed or deleted, please make a permanent DOI’d copy (e.g. in Zenodo) and provide this URL in the manuscript and Data Availability Statement.

------------

We expect to receive your revised manuscript within two weeks. 

*Published Peer Review History*

*Press*

Sincerely,

Taylor

Taylor Hart, PhD, 

Associate Editor

thart@plos.org

PLOS Biology

---

## [Editor Report · Decision Letter 4]

25 Sep 2025

Dear Dr Zhang,

Thank you for the submission of your revised Short Reports "The superficial layers of the primary visual cortex create a saliency map that feeds forward to the parietal cortex" for publication in PLOS Biology. On behalf of my colleagues and the Academic Editor, Frank Tong, I am pleased to say that we can in principle accept your manuscript for publication, provided you address any remaining formatting and reporting issues. These will be detailed in an email you should receive within 2-3 business days from our colleagues in the journal operations team; no action is required from you until then. Please note that we will not be able to formally accept your manuscript and schedule it for publication until you have completed any requested changes.

PRESS

Sincerely, 

Taylor

Taylor Hart, PhD, 

Associate Editor

PLOS Biology

thart@plos.org